# Human macrophages differentially produce specific resolvin or leukotriene signals that depend on bacterial pathogenicity

Oliver Werz[1,2], Jana Gerstmeier[2], Stephania Libreros[1], Xavier De la Rosa[1], Markus Werner[2], Paul C. Norris[1], Nan Chiang[1] & Charles N. Serhan[1]

Proinflammatory eicosanoids (prostaglandins and leukotrienes) and specialized pro-resolving mediators (SPM) are temporally regulated during infections. Here we show that human macrophage phenotypes biosynthesize unique lipid mediator signatures when exposed to pathogenic bacteria. *E. coli* and *S. aureus* each stimulate predominantly proinflammatory 5-lipoxygenase (LOX) and cyclooxygenase pathways (i.e., leukotriene $B_4$ and prostaglandin $E_2$) in M1 macrophages. These pathogens stimulate M2 macrophages to produce SPMs including resolvin D2 (RvD2), RvD5, and maresin-1. *E. coli* activates M2 macrophages to translocate 5-LOX and 15-LOX-1 to different subcellular locales in a $Ca^{2+}$-dependent manner. Neither attenuated nor non-pathogenic *E. coli* mobilize $Ca^{2+}$ or activate LOXs, rather these bacteria stimulate prostaglandin production. RvD5 is more potent than leukotriene $B_4$ at enhancing macrophage phagocytosis. These results indicate that M1 and M2 macrophages respond to pathogenic bacteria differently, producing either leukotrienes or resolvins that further distinguish inflammatory or pro-resolving phenotypes.

[1] Center for Experimental Therapeutics and Reperfusion Injury, Department of Anesthesia, Perioperative and Pain Medicine, Brigham and Women's Hospital and Harvard Medical School, 60 Fenwood Road, BTM 3016, Boston, MA 02115, USA. [2] Department of Pharmaceutical/Medicinal Chemistry, Institute of Pharmacy, Friedrich-Schiller-University Jena, Philosophenweg 14, 07743 Jena, Germany. Correspondence and requests for materials should be addressed to O.W. (email: oliver.werz@uni-jena.de) or to C.N.S. (email: cserhan@bwh.harvard.edu)

Acute inflammation is a host-protective response to infection that can enable elimination of the invading microorganism and facilitate repair of damaged tissue[1]. Initiation and resolution of inflammation are tightly regulated by potent lipid mediators (LM) that can lead to either chronicity or self-resolving inflammation[2,3]. Arachidonic acid (AA)-derived prostaglandins (PG) and leukotrienes (LT) are formed via cyclooxygenase (COX) and 5-lipoxygenase (LOX) pathways and have pivotal functions in initiation of acute inflammation[2]. The temporally produced specialized pro-resolving mediators (SPM) actively stop inflammation to promote resolution of inflammation and tissue regeneration[3]. The SPM superfamily includes lipoxins (LX) biosynthesized from AA, E-series resolvins from eicosapentaenoic acid (EPA), and docosahexaenoic acid (DHA)-derived D-series resolvins, protectins, and maresins[3] that are each produced in humans[3,4].

Chronic inflammation is a central component of numerous widespread diseases, including atherosclerosis, cancer, type 2 diabetes, and Alzheimer's disease that requires therapeutic targeting of the inflammatory response and its resolution[1,5]. As controlled resolution of inflammation is a process that can limit disease chronicity, SPM with dual anti-inflammatory and pro-resolving properties are potential new therapeutics. Along these lines, SPM are host protective in bacterial infections that also lower antibiotic requirements[6,7]. Bacterial infections temporally regulate inflammation-initiating eicosanoids as well as pro-resolving LM during murine peritonitis[6]. Macrophages are central in orchestrating infectious inflammation towards resolution and are actively involved in the clearance of bacteria[8]. Human macrophage phenotypes (M1 and M2) each generate distinct LM signature profiles during polarization. Namely, inflammation-initiating LTB$_4$ and PG are produced in abundance by M1 and pro-resolving LM such as the SPM dominate M2 macrophages[9]. The mechanism for selective LM biosynthesis during infections to produce eicosanoids vs. SPM has not yet been elucidated.

Here, we report that pathogenic bacteria activate human macrophages for differential production of LM from endogenous substrates. Production of LM by macrophage LOX pathways depends on bacterial pathogenicity and on intracellular Ca$^{2+}$ mobilization, while non-pathogenic *Escherichia coli* produces predominantly PG. Our results demonstrate that macrophage polarization dramatically changes the temporal LM biosynthesis

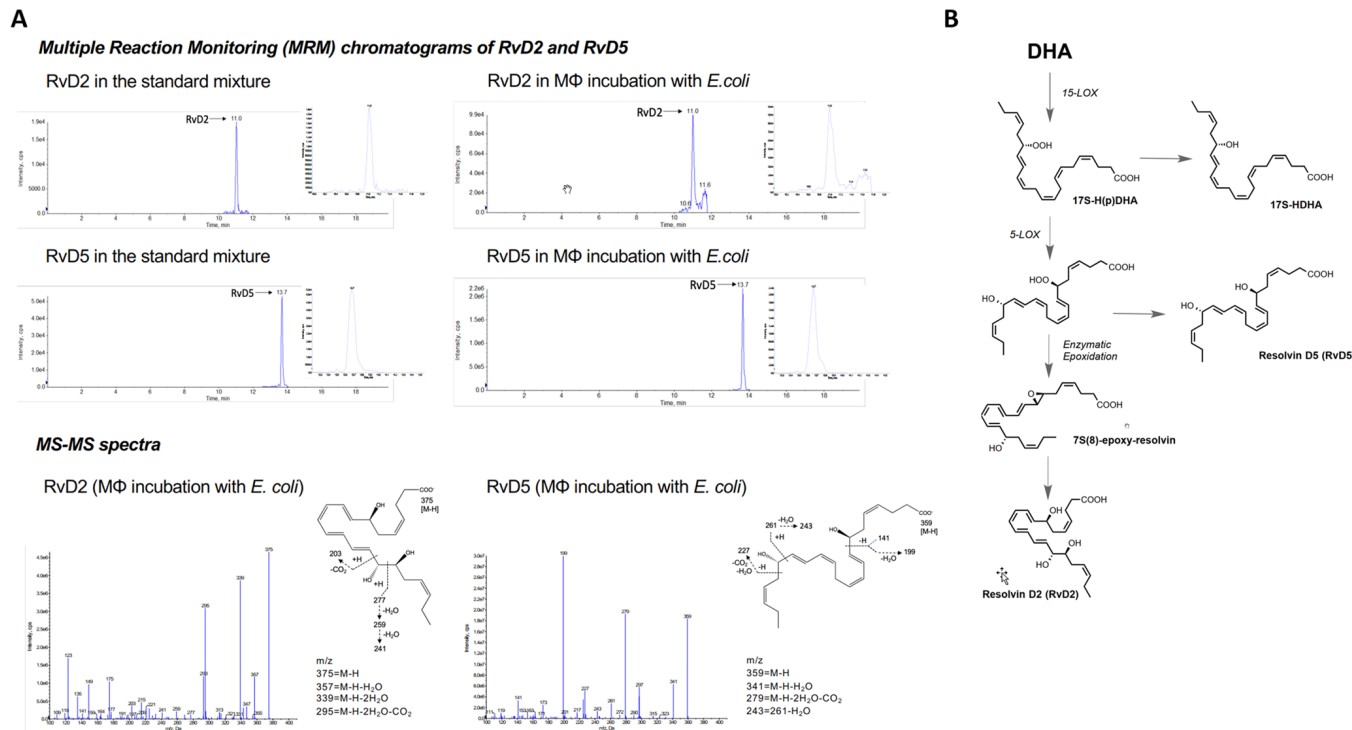

**Fig. 1** a. Human M1 and M2 (5 x 106 cells/ml PBS plus Ca$^{2+}$ and Mg$^{2+}$) plated in 6-well tissue culture plates were incubated for 90 min, 37 °C, pH 7.45 with *E. coli* strain O6:K2:H1; ratio 1:50 Macrophage (MF): *E. coli*. Incubations were stopped by addition of 2 volumes of cold methanol. All incubations were extracted using SPE and subject to targeted LC–MS–MS. (See Supplementary Methods for step-by-step protocol). The >65 targeted LM and pathway markers were profiled. Top panels illustrate screen captures of RvD2 and RvD5 MRM chromatograms obtained from both synthetic resolvins and those obtained from human macrophages. Insets: Enlarged screen captures of the same MRM chromatographs from retention times from 10.0-12.0 mins for RvD2, and 13.2-14.5 mins for RvD5. Bottom panels illustrate screen captures of MS-MS enhanced product ion (EPI) spectra captured from the chromatographic region of RvD2 and RvD5 elution. Blue arrows on Y-axes denote the noise threshold. Insets: Chemical structures and prominent fragmentations. Screen captures of MRM chromatographs and mass spectra of RvD2 and RvD5 obtained from n=7 separate experiments from 7 healthy subjects are provided in a Source Data file. Diagnostic ions are listed and chemical structures shown in Fig. 1a bottom panel insets. The ions used for identification of RvD2 included m/z 375, 357, 339, 295, 277, 259, 241, 203 and these fragmentations are shown in the schematic illustrations in the insets. The ions used for identification of RvD5 included m/z 359, 341, 279, 261, 243, 227, 199, 141 and these fragmentations are shown in the schematic illustrations in the insets.

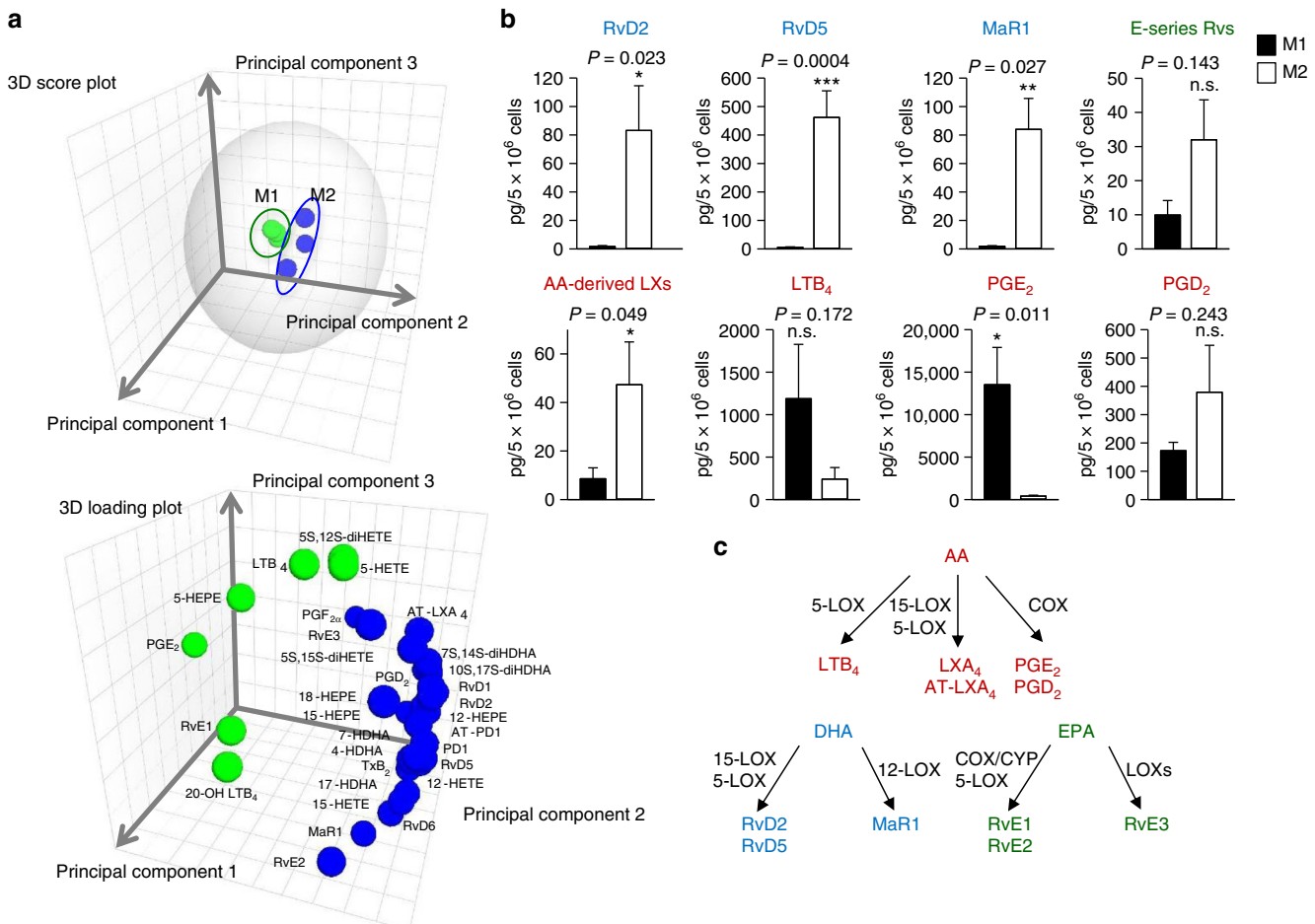

**Fig. 2** M1 and M2 macrophages exposed to *E. coli* produce differential lipid mediators. Human M1 and M2 ($5 \times 10^6$ cells/ml PBS + Ca/Mg) were incubated for 90 min with *E. coli* (O6:K2:H1; ratio, 1:50) at 37 °C. Lipid mediators were isolated by SPE and analyzed by LC–MS–MS. **a** PCA of the lipid mediator profile (SPM, PG, LTB$_4$) of M1 and M2. Upper panel: 3D score plot; lower panel: 3D loading plot. **b** Selected values of lipid mediators formed by M1 and M2 exposed to *E. coli*. Data are given as means ± S.E.M, $n = 7$ (RvD2, RvD5, MaR1, AA-derived LX, LTB$_4$, PGD$_2$, PGE$_2$), $n = 5$ separate donors (E-series resolvins). n.s., not significant ($P > 0.05$); *$P < 0.05$, **$P < 0.01$, ***$P < 0.001$ M1 vs. M2 as determined by two-tailed *t* test. **c** Schematic representation of the respective LM biosynthetic pathways

in response to pathogenic bacteria. Also, we identify a molecular basis for the divergent LM signature profiles between LT vs. SPM biosynthesis that directly affects macrophage host defense.

## Results

**Bacteria elicit differential LM formation in M1 and M2.** To address potential differences in LM profiles by macrophages, we prepared human peripheral blood monocytes differentiated with GM-CSF (6 days) and polarized by lipopolysaccharide (LPS) plus interferon-γ (INFγ) (48 h) to M1, or for M2 with M-CSF (6 days) plus IL-4 (48 h) which are widely used protocols for obtaining these phenotypes in vitro, though they are not necessarily representative of in vivo tissue-specific macrophages[10]. Incubations of these macrophage subtypes with pathogenic *E. coli* (serotype O6:K2:H1) and targeted LM metabololipidomics of supernatants using LC–MS–MS revealed pronounced biosynthesis of 34 distinct LM from the endogenous substrates AA, DHA, and EPA that were each identified (see Supplementary Tables 1 and 2) based on published criteria (i.e., matching chromatographic retention times (RTs), fragmentation patterns, and six characteristic and diagnostic ions)[9,11]. Figure 1a reports the identification of resolvin (Rv)D2 and RvD5 biosynthesized from endogenous substrate from M2 macrophages (see Fig. 1b for biosynthetic scheme). Quantitation of LM using signature ion

pairs obtained via multiple reaction monitoring (MRM) revealed striking differences in the bacteria-elicited LM metabolomes between the M1 and M2 phenotypes (Supplementary Tables 1–3) that was confirmed using unbiased principal component analysis (PCA; Fig. 2a). The M1 and M2 macrophages were each associated with distinct LM (Fig. 2a). M1-derived LM associated with a PG and LT cluster, whereas SPM and their precursors from M2 macrophages gave a specific cluster of pro-resolving LM (Fig. 2a). Flow cytometric analyses of the surface markers CD54 and CD80 for M1, and CD163 and CD206 for M2 confirmed polarization (Supplementary Fig. 1). In response to *E. coli*, M2 produced SPM, specifically RvD2, RvD5, and maresin (MaR)1 (83, 462, and 84 pg/$5 \times 10^6$ cells, respectively) that were produced to a much lesser extent by M1 (1.6, 4.7, and 2.1 pg/$5 \times 10^6$ cells). Of note, M2 produced similar amounts of SPM such as RvD5 vs. LTB$_4$ (Fig. 2b, Supplementary Tables 1 and 2). Levels of E-series resolvins and AA-derived LX were also higher in M2 vs. M1 (Fig. 2b, Supplementary Tables 1 and 2). This contrasts the M1 macrophage LM profile where PG, especially PGE$_2$, and LTB$_4$ clearly dominated the M1 phenotype (Fig. 2b).

Among the >30 LM produced by *E. coli*-challenged M1 and M2 macrophages, only approximately half were identified in the absence of *E. coli* challenge. In these incubations, the LM levels were essentially <10% of *E. coli*-challenged cells (Supplementary

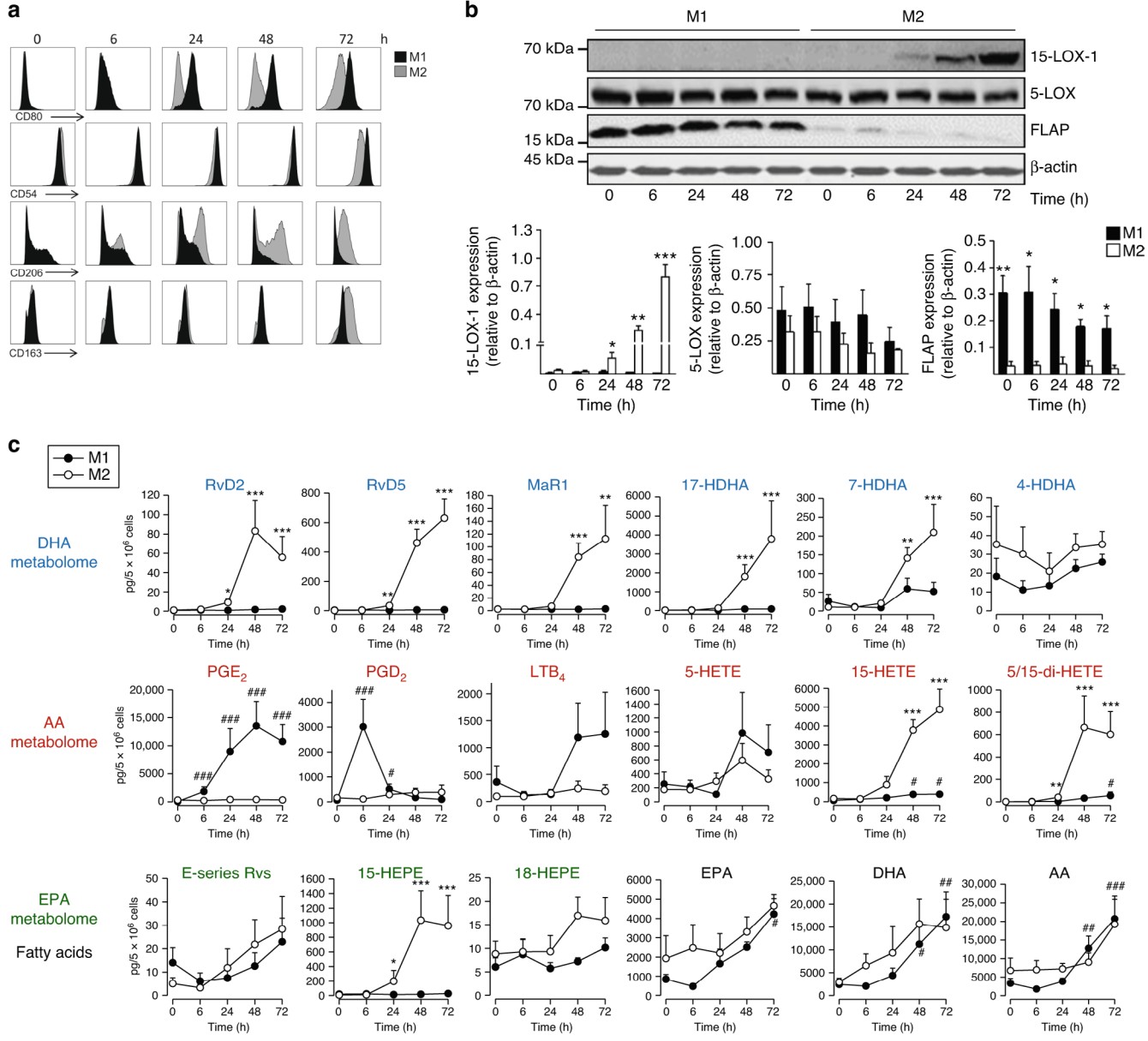

**Fig. 3** Temporal macrophage polarization and lipid mediator pathway induction. Human monocytes were differentiated by GM-CSF or M-CSF (20 ng/ml, each) for 6 days. Cells were either polarized with 100 ng/ml LPS plus 20 ng/ml IFN-γ ($M_{GM-CSF}$) to obtain M1 or with 20 ng/ml IL-4 ($M_{M-CSF}$) to obtain M2. After the indicated times, cells were analyzed for **a** surface expression of polarization markers for M1 (CD54, CD80) and M2 (CD206, CD163) at 0–72 h intervals by flow cytometry; representative histograms from $n = 4$ separate donors, **b** expression of 5-LOX, 15-LOX-1, FLAP, and β-actin by western blot (upper panel), and densitometric analysis thereof (lower panels). Results are given as means ± S.E.M., $n = 4$ separate donors, $*P < 0.05$, $**P < 0.01$, $***P < 0.001$ M1 vs. M2 as determined by two-tailed $t$ test. **c** Lipid mediator formation after incubation of M1 and M2 with $E. coli$ (O6:K2:H1; ratio 1:50) in PBS + Ca/Mg for 90 min at 37 °C after the indicated time points during polarization; formed lipid mediators were isolated by SPE and analyzed by LC–MS–MS. Results are given as means ± S.E.M., $n = 4$ (0 and 6 h), $n = 5$ (24 and 72 h), and $n = 7$ separate donors (48 h). $*,#P < 0.05$; $**,##P < 0.01$; $***,###P < 0.001$ vs. 0 h, data were log transformed for statistical analysis using one-way ANOVA with Bonferroni multiple comparisons test

Tables 1 and 2). Other inflammatory agonists, such as phagocytic stimuli (serum-treated zymosan) or Toll-like receptor (TLR)-chemotactic GPCR challenge (LPS—N-formyl-methionyl-leucyl-phenylalanin (fMLF)) primarily stimulated the formation of PG in M1 and only small amounts of LOX-derived LM as compared to $E. coli$ challenge (Supplementary Fig. 2). Formation of LM by $E. coli$ themselves (incubated alone) was not detected in these experiments. Hence, bacteria can activate human macrophages to preferentially produce pro-inflammatory eicosanoids following M1 polarization and to biosynthesize SPM following M2 polarization.

**Macrophage phenotype-specific LM–SPM pathways and profiles.** The differences in LM profiles of M1 and M2 prompted us to determine the temporal relationships between (i) induction of macrophage phenotype, (ii) LM profiles upon $E. coli$ challenge, and (iii) expression of proteins involved in LM–SPM biosynthesis. Differentiated macrophages (Methods) were polarized and assessed at 0, 6, 24, 48, and 72 h. CD54 and CD80 surface expression was evident after 24 up to 72 h in M1 but not apparent in M2 (Fig. 3a). CD206 increased within 6–24 h in M2, while CD163 was first markedly expressed after 48 h, but was absent in the M1 macrophages (Fig. 3a). In line with the temporal

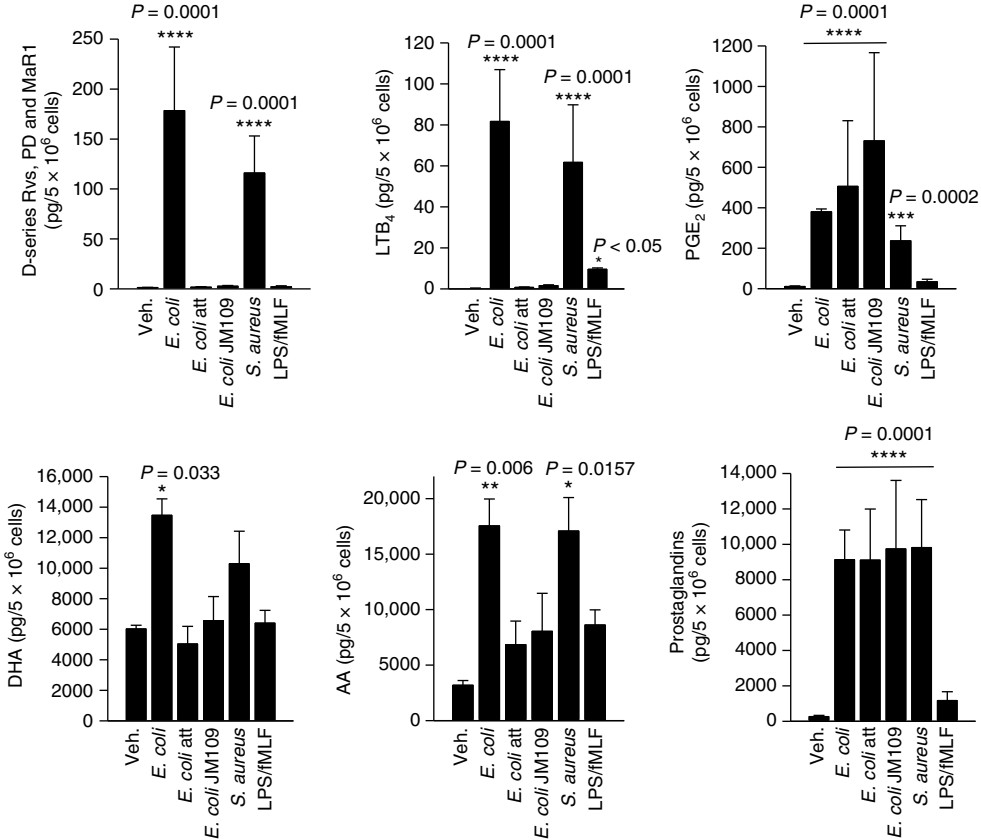

**Fig. 4** M2 macrophages release specific lipid mediators in response to pathogenic bacteria. Human monocyte-derived M2 ($5 \times 10^6$ cells/ml PBS + Ca/Mg) was incubated with *E. coli* (O6:K2:H1), attenuated *E. coli* (O6:K2:H1), non-pathogenic *E. coli* strain JM109, or *S. aureus* (ratio macrophages:bacteria 1:50, each) at 37 °C for 180 min, or with 100 ng/ml LPS plus 100 nM fMLF for 60 min. Lipid mediators were isolated by SPE and analyzed by LC–MS–MS. D-series resolvins include RvD1, RvD2, RvD5, and RvD6, and prostaglandins include $PGD_2$, $PGE_2$, and $PGF_{2\alpha}$. Results are given as means ± S.E.M., $n = 3$ separate donors. *$P < 0.05$, **$P < 0.01$, ***$P < 0.001$, and ****$P < 0.0001$ vs. vehicle control (veh.), data were log transformed for statistical analysis using one-way ANOVA with Dunnett's multiple comparison test

expression of CD163, 15-LOX-1 protein was strongly induced in the M2 at 48 h but not in M1 macrophages (Fig. 3b, Supplementary Fig. 3). While 5-LOX was consistently found in both subtypes, 5-lipoxygenase-activating protein (FLAP) was lower in M2 vs. M1 (Fig. 3b, Supplementary Fig. 3). LM metabolomics of *E. coli*-challenged M2 revealed substantial RvD2, RvD5, and MaR1, as well as 7-hydroxy-DHA (HDHA) and 17-HDHA at 48 and 72 h of polarization (Fig. 3c), correlating with marked 15-LOX-1 expression. A similar pattern was obtained for 15-LOX-1-derived 15-HETE, 15-HEPE, and 5,15-di-HETE, whereas 4-HDHA generation was essentially unchanged. Temporal formation of 5-LOX-derived $LTB_4$ and 5-HETE was clearly different to 15-LOX-1 products and dominated in M1 (Fig. 3c). The 5-LOX in conjunction with 15-LOX-1 produces RvD2, RvD5, and 5,15-di-HETE (Fig. 2c). In M2 macrophages the presence of 15-LOX-1 clearly determined the temporal biosynthesis of these SPM. Note that the formation of E-series resolvins and its precursor 18-HEPE was higher in M2 vs. M1, and apparently did not increase with polarization of the cells (Fig. 3c). $PGD_2$ and $PGE_2$ were produced by M1 cells that were polarized for 24 h, with a peak of $PGD_2$ at 6 h, exceeding its formation in M2, while at 48 or 72 h, formation of $PGD_2$ (but not of $PGE_2$) dominated in M2. Of interest, *E. coli* challenge stimulated release of AA, EPA, and DHA that each increased during polarization without apparent differences between M1 vs. M2. Thus, M1 polarization culminates in proinflammatory $PGE_2$ and $LTB_4$ formation upon bacterial challenge, while acquiring an M2 phenotype is accompanied by

substantial expression of 15-LOX-1 and low expression of FLAP along with a high capacity to biosynthesize SPM.

**LOX-mediated LM formation requires bacterial pathogenicity.** Since *E. coli* stimulated higher levels of LM including SPM vs. individual PAMPs derived from Gram-negative bacteria, e.g., LPS, we questioned whether pathogenicity of bacteria is also conferred by macrophages to produce LM. Compared to pathogenic *E. coli* (O6:K2:H1), the profile and magnitude of LM formed in M2 challenged by *Staphylococcus aureus* was similar (Fig. 4). However, neither attenuated *E. coli* (30 min UV irradiation) nor the non-pathogenic *E. coli* strains JM109 (Fig. 4) or BL21 (Supplementary Fig. 6) elicited LOX-mediated formation of either SPM or $LTB_4$. Also, elevation of DHA and AA release was moderate. Phagocytosis of bacteria can elicit LM formation[12]. Blocking phagocytosis by treatment of M2 with cytochalasin B or D prior to *E. coli* challenge did not prevent the formation of RvD5, $LTB_4$, or $PGE_2$ (data not shown). Surprisingly, COX-derived PG were produced regardless of the pathogenicity of bacteria. LPS plus fMLF slightly stimulated formation of PG and $LTB_4$, and hardly of LOX-derived SPM in M2 (Fig. 4). We conclude that bacterial pathogenicity is required for LOX-mediated LM production in macrophages, while non-pathogenic bacteria stimulate only PG formation, implying differential activation of LOX and COX pathways upon bacteria challenge.

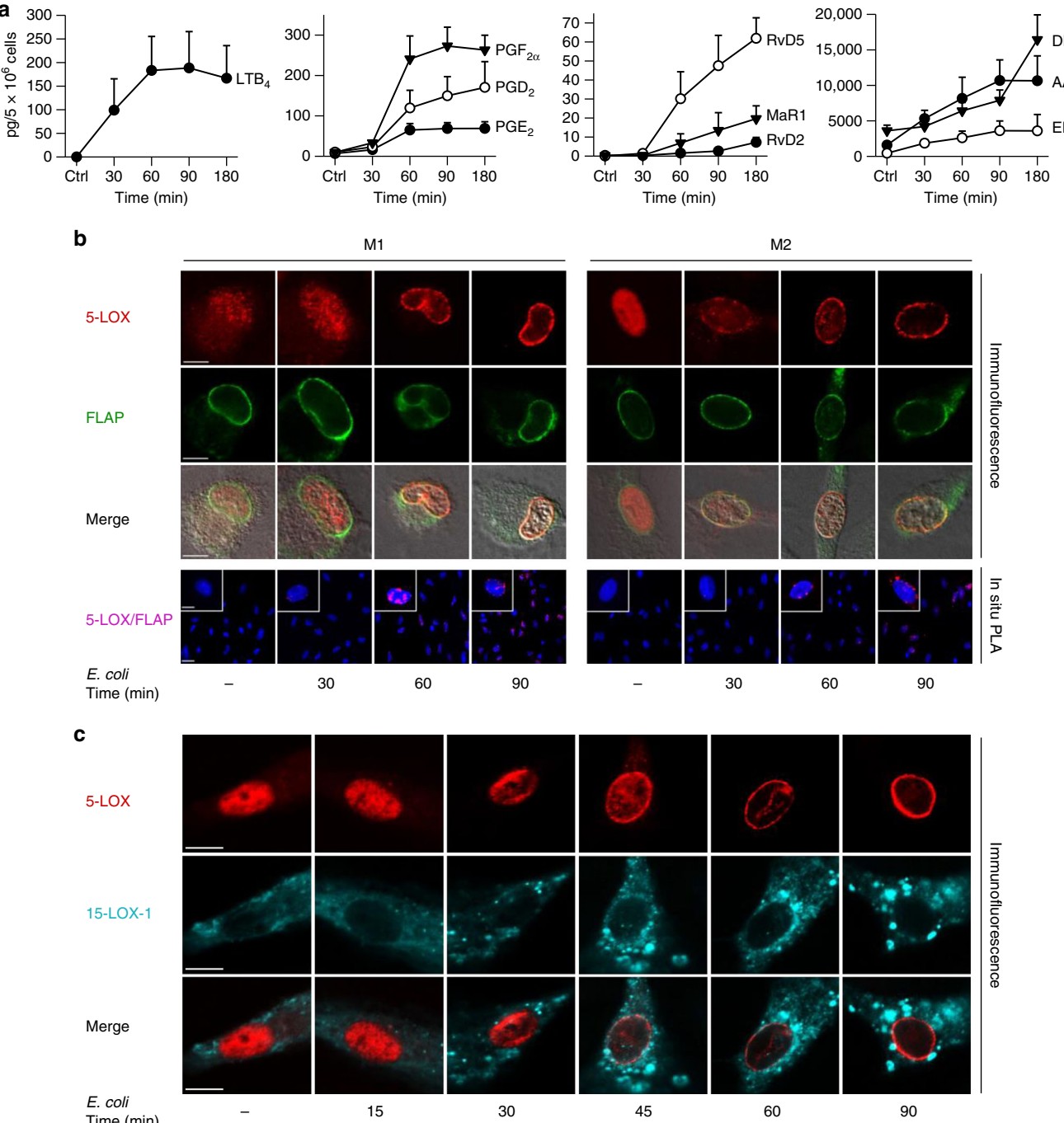

**Fig. 5** Time course of *E. coli*-induced lipid mediator release and lipoxygenase activation. Human M1 or M2 ($5 \times 10^6$ cells/ml PBS + Ca/Mg) was incubated with *E. coli* (O6:K2:H1; ratio = 1:50) at 37 °C for the times indicated. As control (ctrl), cells were incubated at 37 °C for 180 min (**a**) or 90 min (**b**, **c**) without *E. coli*. **a** Formed lipid mediators in M2 were isolated by SPE and analyzed by LC–MS–MS. Data are given as means ± S.E.M., *n* = 3 separate donors. **b**, **c** Subcellular redistribution of 5-LOX and FLAP in M1 and M2 (**b**) or 5-LOX and 15-LOX-1 in M2 (**c**). After exposure to *E. coli* for the indicated times, cells were fixed, permeabilized, and incubated with antibodies against 5-LOX (red), 15-LOX-1 (cyan-blue), and FLAP (green); scale bars = 5 μm. **b** In situ interaction of 5-LOX and FLAP in M1 and M2 (lower panel); scale bars = 5 μm (insets) and 10 μm (overview). Proximity ligation assay (PLA) was performed after exposure of cells to *E. coli* after the indicated times. DAPI (blue) was used to stain the nucleus and in situ PLA signals (magenta dots) visualize 5-LOX/FLAP interaction. Results shown for one single cell are representative for ~100 individual cells analyzed in *n* = 3 independent experiments (separate donors)

**Temporal LM–SPM production and LOX translocation**. During bacterial infections in mice, inflammation-initiating LT and PG are rapidly formed, while SPM generation is delayed and peaks at the resolution phase[6,13]. We asked if temporal LM–SPM formation would also occur in human macrophages exposed to pathogenic *E. coli* (O6:K2:H1) and correlate with activation of respective LOX (5-LOX vs. 15-LOX). Here, both $LTB_4$ and PG

were rapidly formed, reaching half-maximal levels ($t_{50\%}$) after 30–50 min, with plateau at 60–90 min (Fig. 5a). In contrast, RvD2, RvD5, and MaR1 first appeared after 60 min and continuously increased up to 180 min.

The $Ca^{2+}$ ionophores (e.g., A23187), GPCR ligands (e.g., fMLF and C5a), or zymosan cause 5-LOX translocation to the nuclear membrane and interaction with FLAP to activate LM formation

in leukocytes[14]. Similarly, 15-LOX-1 redistributes to specific cellular membranes in eosinophils or monocytes upon stimulation accompanied by LM biosynthesis[15]. The subcellular locales and trafficking routes of LOXs in human M1 or M2 were not known. Immunofluorescence microscopy revealed predominant intranuclear 5-LOX in M2 but also partially in the cytosol of M1

(Fig. 5b). Exposure of both M1 and M2 to pathogenic *E. coli* caused 5-LOX translocation to the nuclear membrane within 30–60 min, co-localizing with FLAP (Fig. 5b) correlating with LTB$_4$ formation (Fig. 5a). In human monocytes and neutrophils, 5-LOX interacts with FLAP upon stimulation with Ca$^{2+}$ ionophore[16]. Using proximity ligation assay (PLA), we found

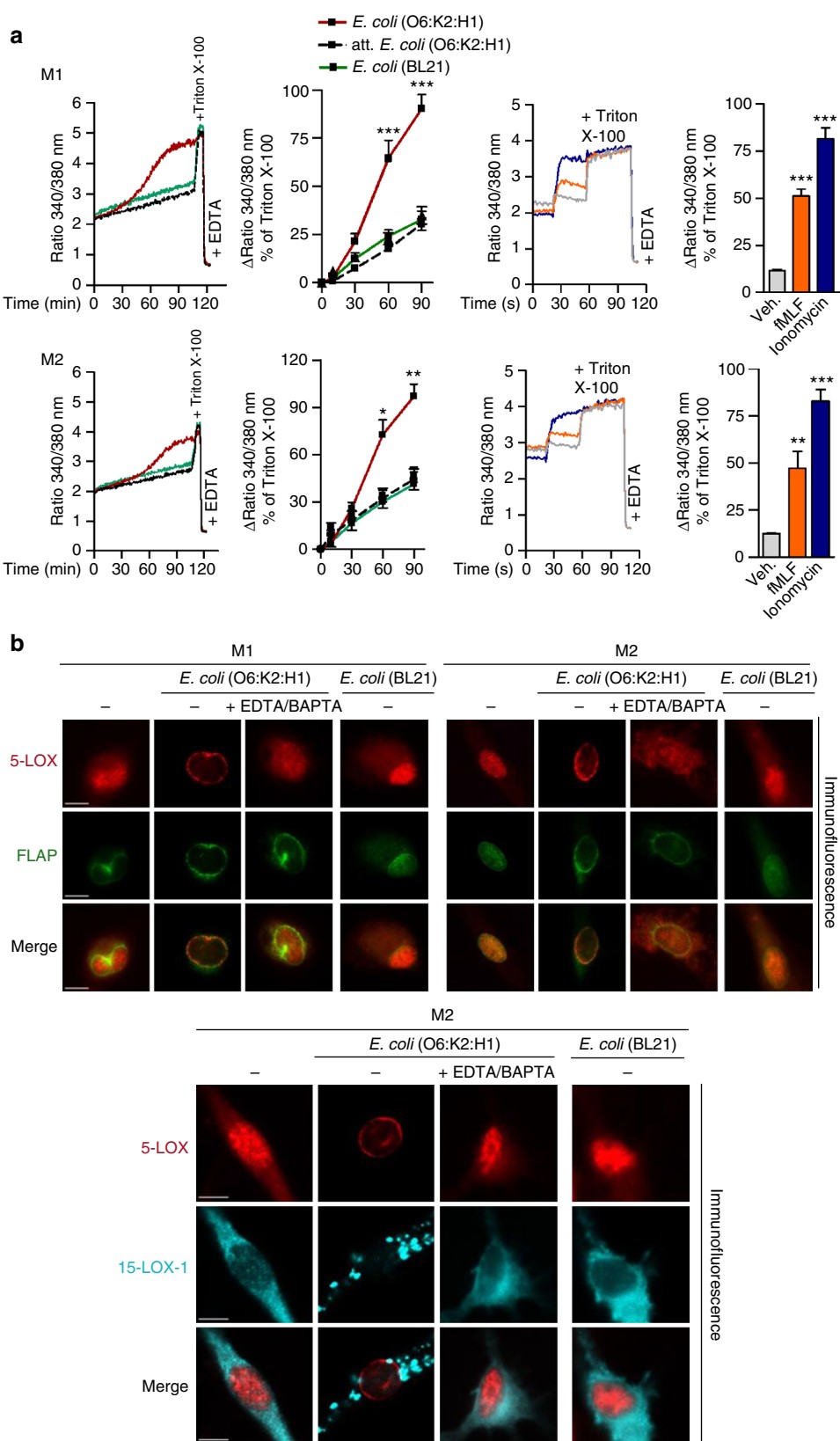

perinuclear 5-LOX/FLAP complexes in both M1 and M2 macrophages at 30 min after exposure to *E. coli*, with maximal signals at 90 min and slightly less intensities in M2 (Fig. 5b). The 15-LOX-1 was predominantly cytosolic in M2 cells and *E. coli* caused punctuated accumulation of 15-LOX-1 within the cytosol, while 5-LOX within the same cells localized at the nuclear membrane (Fig. 5c). LPS–fMLF-treatment of M1 or M2 macrophages did not evoke LOX translocation (Supplementary Fig. 4). The localization of 15-LOX-1 in M1 cells could not be assessed, likely because of low 15-LOX-1 expression (Fig. 3b). Thus, *E. coli* activates temporal translocation of 5-LOX and 15-LOX-1 to different subcellular locales in macrophages, correlating with the differential time course of eicosanoid vs. SPM formation.

Next, we found that a widely used FLAP inhibitor (MK886 at 100 nM) prevented *E. coli*-induced 5-LOX/FLAP complex assembly (Supplementary Fig. 5) as well as reduced the formation of 5-LOX-dependent LM from AA (i.e., LTB$_4$ and its trans-isomers, 5-HETE, 5,15-diHETE, as well as LXA$_4$) in M1 or M2, as expected (Supplementary Fig. 5). Complexes of 15-LOX-1 with 5-LOX or 15-LOX-1 with FLAP were not detectable by PLA (not shown). MK886 did not inhibit formation of SPM from DHA in M2 (e.g., RvD5, protectin (PD)1, MaR1), nor production of 15-LOX-1-derived 15-HETE, 15-HEPE, or 17-HDHA pathway markers (Supplementary Fig. 5). These results suggest that FLAP was not required for biosynthesis of DHA-derived SPM including RvD5, MaR1, and PD1. MaR1 and 7s,14S-diHDHA from the maresin pathway are biosynthesized via initial oxygenation by the human 12-LOX and their biosynthesis is independent of 5-LOX[17,18]. Also, PD1 biosynthesis is initiated via the 15-LOX-1 and does not require 5-LOX[3,19]. Thus, FLAP is not involved in the biosynthesis of PD1, MaR1, or 7S, 14S-diHDHA and MK886 did not affect the production of their pathway markers (i.e., 17-HDHA and 14-HDHA).

**Role of Ca$^{2+}$ for bacterial LOX activation**. Translocation and binding of 5-LOX and 15-LOX-1 to membranes and related LM formation depends on Ca$^{2+}$ ions[14,15]. Compared to fMLF or ionomycin, exposure of M1 and M2 to pathogenic *E. coli* (O6:K2:H1) caused delayed elevation of intracellular Ca$^{2+}$ concentrations ([Ca$^{2+}$]$_i$) (Fig. 6a), correlating with temporal LOX translocation (Fig. 5b, c). Non-pathogenic *E. coli* (attenuated or the BL21 strain) failed to elevate [Ca$^{2+}$]$_i$ (Fig. 6a) and to induce 5-LOX/15-LOX-1 translocation (Fig. 6b) in macrophages. Removal of Ca$^{2+}$ (using 0.5 mM EDTA and 20 µM BAPTA/AM) impeded translocation of 5-LOX and 15-LOX-1 in response to pathogenic *E. coli* (Fig. 6b) and also abolished LOX-dependent LM formation, while PG were still formed to some extent (Supplementary Fig. 6). Thus, *E. coli*-induced LOX activation is Ca$^{2+}$ dependent and correlates to the ability of bacteria to elevate [Ca$^{2+}$]$_i$.

**RvD5 vs. LTB$_4$ function in bacterial phagocytosis**. The differences in the *E. coli*-induced LM profiles in M1 and M2 support

different roles of these macrophage phenotypes in initiation or propagating inflammation (M1) and resolution (M2). While M1 exposed to *E. coli* produced 250-fold higher levels of pro-inflammatory LTB$_4$ vs. pro-resolving RvD5, in M2 the formation of RvD5 dominated over LTB$_4$ (Supplementary Table 1). To address the bacterial killing capacity of M1 and M2, the macrophages were incubated with *E. coli* (ratio 1:50) for 2 h and bacterial titers were determined. The live bacterial titers, i.e. colony forming units (CFU), recovered from M2 (43.8 ± 8.1 × 10$^6$) were ~40% lower than CFU from M1 (74.8 ± 13.6 × 10$^6$; $P < 0.05$), indicating that M2 had higher bacterial killing capacity than the M1 macrophages. Next, we examined the actions of RvD5 vs. LTB$_4$ in regulating phagocytosis. In M1 macrophages, RvD5 (10 nM) enhanced phagocytosis of fluorescent-labeled *E. coli* by ~70% compared to *E. coli* alone, exceeding the effect of LTB$_4$ (10 nM) (Fig. 7a, b). By comparison, neither RvD5 nor LTB$_4$ increased M2 macrophage phagocytosis of *E. coli*. These potencies of RvD5 and LTB$_4$ correlated with the expression levels of their high-affinity GPCRs (i.e., RvD1 receptor (DRV1/GPR32) for RvD5 and LTB$_4$ receptor (BLT1) for LTB$_4$) assessed by flow cytometry. Thus, DRV1 was strongly expressed on M1 and to a minor extent on M2, whereas BLT1 was only moderately expressed on both macrophage phenotypes (Fig. 7c). These results demonstrated that RvD5 is a potent stimulator of phagocytosis with M1 macrophages that positively correlates with surface expression of DRV1.

## Discussion

In the present report, we demonstrate that pathogenic bacteria are able to evoke phenotype-specific LM signatures from M1 and M2 macrophages produced from endogenous substrates. These results with human macrophages are consistent with temporal formation of distinct LM obtained during bacterial infections in mice in vivo where different LM are biosynthesized during initiation (such as PG and LT), while SPM (resolvins, protectins, and maresins) are produced and function during resolution of inflammation to accelerate the termination of the response[6,13]. Inflammation-initiating 5-LOX and COX products were produced by pathogenic *E. coli* or *S. aureus* predominantly in M1, whereas in M2 the pathogens activated an abundant production of bioactive SPM from endogenous substrates via 15-LOX-1. Of interest, only pathogenic bacteria stimulated LOX-mediated LM–SPM production along with Ca$^{2+}$-dependent translocation of 5-LOX and 15-LOX-1. Importantly, these actions were not mimicked by the isolated bacterial components namely LPS or fMLF. Hence, divergent temporal and spatial cellular regulation of LOX pathways in M1 and M2 evoke phenotype-specific LM profiles after bacteria challenge. These distinct LM signal profiles might constitute crucial factors that can help to determine different functions of pro-inflammatory M1 macrophages vs. pro-resolving M2 phenotype[20]. These signature profiles of LM–SPM that proved to be dependent on the pathogenicity of *E. coli* or *S.*

**Fig. 6** *E. coli* elevates [Ca$^{2+}$]$_i$ in M1 and M2 and Ca$^{2+}$ is required for LOX translocation. **a** Measurement of [Ca$^{2+}$]$_i$. Fura-2/AM-loaded M1 or M2 in PBS containing 1 mM Ca$^{2+}$ was treated with 1 µM fMLF, 2 µM ionomycin or vehicle (0.1% DMSO) for 10 min or with *E. coli* (O6:K2:H1), attenuated *E. coli* (O6:K2:H1), or non-pathogenic *E. coli* strain BL21 (ratio 1:50, each), or PBS + Ca/Mg as vehicle at 37 °C for up to 90 min. The ratio of absorbance at 340 vs. 380 nm reflecting [Ca$^{2+}$]$_i$ is given as percentage of cells that were lysed with Triton X-100 (=100% control). Data are given as means ± S.E.M., $n = 4$ separate donors. *$P < 0.05$; **$P < 0.01$; ***$P < 0.001$ vs. $t = 0$ s or min, one-way ANOVA with Tukey's multiple comparison test. **b** Subcellular redistribution of 5-LOX and FLAP in M1 and M2 (upper panel) or 5-LOX and 15-LOX-1 in M2 (lower panel) in the presence or absence of Ca$^{2+}$. M1 or M2 (1 × 10$^6$ cells/ml) was incubated with *E. coli* (O6:K2:H1) or non-pathogenic *E. coli* strain BL21 (ratio 1:50, each) in PBS + Ca/Mg or PBS plus 0.5 mM EDTA and 20 µM BAPTA/AM at 37 °C as indicated. Cells were then fixed after 90 min, permeabilized, and incubated with antibodies against 5-LOX (red), 15-LOX-1 (cyan-blue), and FLAP (green); scale bars = 5 µm. Results shown for one single cell are representative for ~100 individual cells analyzed in $n = 3$ independent experiments (separate donors)

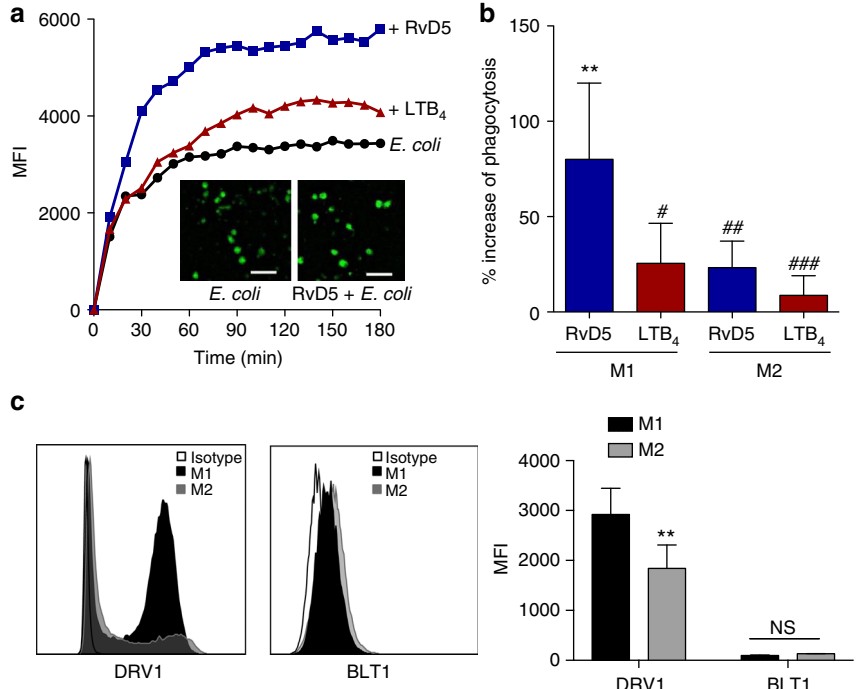

**Fig. 7** RvD5 stimulates phagocytosis of *E. coli* in M1 vs. M2 human macrophages. **a** Effects of RvD5 and $LTB_4$ on phagocytosis of *E. coli* by M1. Results are mean fluorescence of four fields/well from one representative donor (inset, representative fluorescent image); scale bar, 50 μm. **b** % Increase of phagocytosis above vehicle control (*E. coli* alone) at 180 min; mean ± S.E.M. of $n = 4$ separate donors. $**P = 0.001$ vs. vehicle, $\#P = 0.028$, $\#\#P = 0.021$, $\#\#\#P = 0.003$ vs. M1 incubated with RvD5 using one-way ANOVA with Bonferroni multiple comparison test. **c** Surface expression of DRV1 (GPR32) and BLT1 in M1 and M2 by flow cytometry. Left panel: representative overlays histogram plots gated on M1 or M2 for DRV1 or BLT1 surface expression. Right panel: quantification of expression of DRV1 and BLT1 in M1 and M2. Results are obtained from $n = 6$ separate donors, data are means ± S.E.M. $**P = 0.0034$ and NS = 0.5838, M1 vs. M2 as determined by two-tailed *t* test

*aureus* could each contribute to the regulation of infections in humans given the potent actions of these mediators in vivo[1–3].

During polarization, M1 and M2 acquired the sequential expression of heterogeneous LM pathways, where the inflammation-initiating COX-2- and 5-LOX-mediated pathways were rapidly induced and predominated in M1. The M1 cells express more FLAP than M2. The 15-LOX-1 is essentially absent in M1, and thus SPM (i.e., RvD5, 7S,14S-diHDHA, and Mar1) formation is very low just above the detection limits with levels that were ~100-fold lower than in the M2 cells. This made it not plausible to accurately assess a role for FLAP (using MK886) in SPM formation in M1 cells since they did not produce appreciable amounts of SPM. Importantly, the pro-resolving M2 cells expressed much higher 15-LOX-1 that can preferably produce SPM. The capacity to release AA, DHA, and EPA in response to *E. coli* was enhanced during polarization to M1 and M2. Thus, the differential LM production in M1 vs. M2 is likely not due to limitations in substrate release. Instead, strikingly higher 15-LOX-1 protein expression in M2 above that of the M1 cells appears to account for the greater SPM production by M2 cells. Whereas higher FLAP levels may support superior $LTB_4$ formation in M1 than SPM. Moreover, it is likely that subcellular substrate access for 5-LOX and/or 15-LOX-1 at different intracellular sites within the two macrophage phenotypes also contributes to the unique cell type-specific LM signature profiles they produce. Among SPM in self-resolving murine peritoneal *E. coli* infections, RvD5 is most abundant in the exudates[6]. Of note, RvD5 was the dominant SPM formed from *E. coli*-challenged human M2 macrophages. Here, RvD5 showed 100-fold higher amounts in M2 than in M1 cells and exceeding the amounts of $LTB_4$ produced by the M2 macrophages. These divergent LM signatures are in line with those reported for human macrophages during polarization and

after efferocytosis of apoptotic neutrophils[9], and they are consistent with the well-appreciated pro-inflammatory role of the M1 macrophages and the functions of M2 in helping to orchestrate the resolution of inflammation and tissue repair[8,20]. Recently, in atherosclerosis imbalances between SPM and LT amounts were reported, where SPM levels in histologically defined stable regions of human carotid atherosclerotic plaques were higher than in vulnerable plaque regions[21].

In line with the temporal LM production during bacterial infections in vivo[6], pro-inflammatory $LTB_4$ and $PGE_2$ that promote phagocyte recruitment and edema in the early phase of inflammation were rapidly formed in *E. coli*-challenged macrophages, while SPM (e.g., RvD5) formation was delayed, congruent with the property to stimulate bacterial clearance during resolution of inflammation at later stages. Indeed, further support for this comes with the finding that RvD5 strongly enhanced phagocytosis of *E. coli* by M1 that abundantly express DRV1 as compared to $LTB_4$ and moderate BLT1 expression. This suggests that $LTB_4$ plays a crucial role in neutrophil chemotaxis at the initiation phase of inflammation[22], and that RvD5 produced during the resolution phase functions as a major phagocytic signal for macrophages.

Despite the well-recognized fundamental and perpetual combat of macrophages with bacteria[23,24], consequent induction of LM–SPM formation in bacteria-challenged human macrophages remained elusive. The present results show that the marked activation of LM pathways in human macrophages via pathogenic bacteria, a pathophysiological relevant condition, leads to pronounced LM levels from endogenous substrates. Results obtained with isolated primary human cells that addressed mechanistic questions on 5-LOX-mediated LM biosynthesis have applied exogenous agents that evoke almost immediate (within seconds)

robust cell activation such as $Ca^{2+}$ ionophore (i.e., A23187), fMLF, platelet-activating factor, or C5a[14]. Relevant microbes that markedly induce 15-LOX-1 product biosynthesis from endogenous substrates in isolated cells have not been reported earlier. Here, *E. coli* evoked significant higher LM levels in M2 as compared to serum-treated zymosan or LPS–fMLF, agents that are commonly used in studies investigating LM biosynthesis in macrophages. Surprisingly, pathogenicity was a requisite for LOX-dependent LM formation, while PG production was essentially equally produced in response to non-pathogenic or pathogenic bacteria. These new findings implicate divergent factors and activation mechanisms of LOX pathways. LPS, a membrane component of Gram-negative bacteria, potently activates macrophages for secretion of cytokines (e.g., TNF-α, IL-1β, IL-6), nitric oxide and pro-inflammatory LM via CD14 and TLR4[25]. Interestingly, activation of another TLR signaling route, i.e., TLR7 that recognizes viral ssRNA and damaged self-RNA, stimulated DHA-derived SPM production in macrophages and promoted resolution of airway inflammation[26]. In our present results, LPS (plus fMLF) did not mimic *E. coli*-induced SPM formation or translocation of 5-LOX or 15-LOX-1. These agonists LPS and fMLF mainly stimulated formation of PG in M1, the pattern observed for non-pathogenic bacteria. Phagocytosis-related activation is an unlikely explanation since cytochalasins B and D (inhibitors of phagocytosis) did not block *E. coli*-activated LM formation. Thus, PAMP recognition alone by human macrophages is not sufficient to stimulate the biosynthesis of SPM.

Upon cell activation, 5-LOX and 15-LOX-1 translocate from soluble to membranous compartments to receive substrate(s) for LM formation[14,15]. Apart from substrate utilization of AA, and EPA, we cannot fully explain all of the effects of MK886 on such complex LM networks. Some actions of MK886 may be off target in these cell types. Also, $cPLA_2$ (that releases AA and EPA) may participate within the 5-LOX/FLAP complex which is disrupted by MK886 and might thus negatively affect $cPLA_2$. It is possible that $LTB_4$ or 5-HETE may act as autocrine enhancers for further AA and EPA release that would be abolished when MK886 blocks $LTB_4$ and 5-HETE biosynthesis. The subcellular localization and trafficking of 5-LOX or 15-LOX-1 in M1 and M2 was yet unknown. Here, our experiments revealed distinct subcellular distribution patterns of these LOXs in M1 and M2. These results demonstrate for the first time that bacteria causes 5-LOX nuclear translocation and interaction with FLAP. Unexpectedly, while 5-LOX moved to the nuclear envelope, 15-LOX-1 accumulated in the cytosol of M2 after *E. coli* challenge, excluding co-localization of 15-LOX-1 with 5-LOX and FLAP. The substrate supply at these different locales may vary, where 15-LOX-1 is preferentially provided with DHA in M2 cells and 5-LOX with AA in the M1 cells. RvD5 formation requires first 15-LOX-1-mediated DHA lipoxygenation to produce 17-H(p)DHA that is then converted by 5-LOX to RvD5[3]. The apparent lack of co-localization of these LOXs implies that the 15-LOX-1-formed 17-HDHA is effectively shuttled to 5-LOX, seemingly independent of FLAP as MK886 did not block RvD5 formation in the M2 cells. It is possible that in M2 macrophages the RvD5 is produced via 15-LOX-1 alone to insert both molecules of molecular oxygen without the need of 5-LOX or FLAP. Formation of the two precursors of RvD5 (7S,17S-diHDHA), namely 7S-HDHA and 17S-HDHA, was not directly inhibited by MK886 in M2 cells suggesting that the oxygenations at 7- and 17-positions in these cells from endogenous substrate leading to RvD5 appear to be 5-LOX/FLAP independent. Along these lines, 15-LOX-1 is able to catalyze all steps in the biosynthesis of $LXA_4$, without the need of 5-LOX[27]. In this context, it was shown that activated cytoplasmic 5-LOX in macrophages, without access to FLAP favors SPM biosynthesis over the production of $LTB_4$[28], suggesting a limited

role for FLAP in leukotriene biosynthesis. 5-LOX and 15-LOX-1 are known to require $Ca^{2+}$ for membrane translocation and full enzymatic activity[14,15]. Pathogenic *E. coli* elevated $[Ca^{2+}]_i$ in M1 and M2 positively correlating with $Ca^{2+}$-dependent 5-LOX/15-LOX-1 translocation and LOX-mediated LM formation that contrasts the non-pathogenic bacteria.

Together, these results demonstrate that pathogenic bacteria are able to evoke the biosynthesis of specific LM profiles from endogenous substrates via activation of LOX pathways associated with macrophage phenotypes. Notably, these different LM signature profiles contribute to the functions of pro-inflammatory M1. These are distinct LM profiles than those from the pro-resolving M2 phenotype that produce specific SPM with pathophysiologic relevant bacteria.

## Methods

**Cell isolation and polarization of macrophages.** Human monocytes were isolated from de-identified leukopacks obtained from Children's Hospital Blood Bank (Boston, MA) or the Institute of Transfusion Medicine, University Hospital Jena, Germany, with the use of Ficoll-Histopaque 1077-1 (Sigma-Aldrich, St. Louis, MO). Blood was obtained from healthy human volunteers giving informed consent under protocol #1999-P-001297 approved by the Partners Human Research Committee. The protocols for experiments with macrophages were approved by the ethical commission of the Friedrich-Schiller-University Jena. All methods were performed in accordance with the relevant guidelines and regulations. For differentiation and polarization towards M1 and M2, published criteria were used[9]. Briefly, M1 was produced by incubating isolated monocytes with GM-CSF (20 ng/ml) for 6 days in RMPI 1640 (Life Technologies, Carlsbad, CA) supplemented with 10% fetal bovine serum (Invitrogen, Grand Island, NY), 2 mmol/l L-glutamine (Lonza, Basel, Switzerland), and penicillin–streptomycin (Lonza), followed by LPS (100 ng/ml) plus INF-γ (20 ng/ml) treatment for the indicated times (routinely 48 h). M2 was obtained by incubating monocytes with 20 ng/ml M-CSF for 6 days followed by polarization with 20 ng/ml IL-4 for the indicated times (routinely 48 h).

**Flow cytometry.** Fluorescent staining for flow cytometric analysis was performed in M1 or M2 in FACS buffer (PBS with 1% bovine serum albumin and 0.1% sodium azide). Fc receptor-mediated non-specific antibody binding was blocked by using human TruStain FcX solution (Biolegend, San Diego, CA). Macrophages were stained for 30 min at 4 °C. The following antibodies were used: anti-human APC CD54 (HA58; Cat. no. 559771; 15 μl/$10^6$ cells in 100 μl), anti-human FITC CD206 (19.2; Cat. no. 551135; 10 μl/$10^6$ cells in 100 μl) (BD Bioscience, San Jose, CA), anti-human PE CD80 (2D10; Cat. no. 305208; 10 μl/$10^6$ cells in 100 μl), and anti-human PerCP/Cy5.5 CD163 (RM3/1; Cat. no. 326512; 15 μl/$10^6$ cells in 100 μl) (Biolegend, San Diego, CA). For surface expression of DRV1 and BLT1, polarized M1 and M2 were stained with PE anti-human BLT1 (203/14F11; Cat. no. 552836; 15 μl/$10^6$ cells in 100 μl; BD Bioscience) or rabbit anti-human DRV1 (GPR32) antibody (Cat. no. GTX108119; 2 μl/$10^6$ cells in 100 μl; GeneTex, Irvine, CA), followed by non-immune rabbit IgG for 30 min. Macrophages (M1 or M2) were analyzed using FACSCanto II (BD Bioscience) flow cytometer, and data analyzed using FlowJo X Software.

**Macrophage incubation and LM metabololipidomics.** Macrophages were routinely incubated at $5 \times 10^6$ cells/ml of PBS with $Ca^{2+}$ and $Mg^{2+}$ (PBS + Ca/Mg). In experiments where macrophages were treated in the presence of 0.5 mM EDTA and 20 μM BAPTA/AM, cells were incubated in PBS. Freshly grown pathogenic *E. coli* (serotype O6:K2:H1, except stated otherwise) or non-pathogenic *E. coli* JM-109 or *E. coli* BL21 strains were added at a ratio of 1:50 (macrophages:*E. coli*), and incubated at 37 °C for the indicated times. The supernatants were transferred to 2 ml of ice-cold methanol containing the deuterium-labeled internal standards d$_8$-5S-HETE, d$_4$-LTB$_4$, d$_5$-LXA$_4$, d$_5$-RvD2, and d$_4$-PGE$_2$ (500 pg, each) to facilitate quantification and sample recovery. Samples were kept at −20 °C for 60 min to allow protein precipitation and then centrifuged ($1200 \times g$, 4 °C, 10 min). Solid-phase C18 cartridges were equilibrated with 6 ml methanol before the addition of 6 ml $H_2O$. Next, 9 ml acidified $H_2O$ (pH 3.5, HCl) was added to the samples, and loaded onto the conditioned C18 columns that were washed once with 6 ml $H_2O$, followed by 6 ml hexane. The products were eluted with 6 ml of methyl formate. Samples were brought to dryness using an evaporation system (TurboVap LV, Biotage) and immediately suspended in methanol–water (50/50 vol/vol) for LC–MS–MS automated injections.

The LC–MS–MS system employed was equipped with a Shimadzu LC-20AD HPLC and a Shimadzu SIL-20AC autoinjector (Shimadzu, Kyoto, Japan), coupled with a QTrap 5500 (ABSciex, Framingham, MA). An Eclipse Plus C18 column ($100 \times 4.6$ mm $\times 1.8$ μm; Agilent) was kept in a column oven maintained at 50 °C (ThermaSphere TS-130; Phenomenex, Torrance, CA), and LM were eluted with a mobile phase consisting of methanol–water–acetic acid of 55:45:0.01 (vol/vol/vol)

that was ramped to 85:15:0.01 (vol/vol/vol) over 10 min and then to 98:2:0.01 (vol/vol/vol) for the next 8 min. This was subsequently maintained at 98:2:0.01 (vol/vol/vol) for 2 min, and the flow rate was maintained at 0.4 ml/min. The QTrap 5500 was operated in negative ionization mode using scheduled MRM coupled with information-dependent acquisition (IDA) and an enhanced product ion scan. The scheduled MRM window was 90 s, and each LM parameter was optimized individually.

To monitor each LM and their respective pathways, an MRM method was used with diagnostic ion fragments and identification using recently published criteria[9], including matching RT to synthetic and authentic materials and at least six diagnostic ions for each LM. Calibration curves were obtained for each using authentic compound mixtures and deuterium-labeled LM at 3.12, 6.25, 12.5, 25, 50, 100, and 200 pg (e.g., $d_8$-5S-HETE, $d_4$-LTB$_4$, $d_5$-LXA$_4$, and $d_5$-RvD$_2$). Linear calibration curves were obtained for each, which gave $r^2$ values of 0.98–0.99. Internal standard recoveries, interference of the matrix, and limit of detection (range of 20–220 fg for the QTrap 5500 in tissue and in biological matrix) were determined.

**Principal component analysis.** PCA was performed using SIMCA 13.0.3 software (MKS Data Analytics Solution Umea, Sweden) following mean centering and unit variance scaling of LM amounts. PCA serves as an unbiased, multivariate projection designed to identify the systematic variation in a data matrix (the overall bioactive LM profile of each sample) with lower dimensional plane using score plots and loading plots. The score plot shows the systematic clusters among the observations (closer plots presenting higher similarity in the data matrix). Loading plots describe the magnitude and the manner (positive or negative correlation) in which the measured LM/SPM contribute to the cluster separation in the score plot. For graphic illustrations, direct screen captures of MRM chromatographs and MS-MS spectra from LC-MS-MS were obtained from the chromatographic regions for RvD2 and RvD5 based on retention times of synthetic standards (Cayman Chemicals) and published criteria (Hong et al. 2003; PMID:12590139). Spectra were obtained using Centroid Mode in Analyst software (Version 1.6.2; AB Sciex, Framingham, Massachusetts).

**SDS-PAGE and western blot.** Cell lysates of macrophages, corresponding to $3 \times 10^6$ cells, were separated on 10% (5-LOX, 15-LOX-1) and 16% (FLAP) polyacrylamide gels, and blotted onto nitrocellulose membranes (Hybond ECL, GE Healthcare, Freiburg, Germany). The membranes were incubated with the following primary antibodies: mouse monoclonal anti-15-LOX-1, 1:500 (ab119774, Abcam, Cambridge, UK); rabbit polyclonal anti-5-LOX, 1:500 (1550 AK6, provided by Dr. Olof Radmark, Karolinska Institutet, Stockholm, Sweden); rabbit polyclonal anti-FLAP, 0.1 μg/ml (ab85227, Abcam, Cambridge, UK); rabbit anti-β-actin, 1:1000 (4967S, Cell Signaling, Danvers, MA). Immunoreactive bands were stained with IRDye 800CW Goat anti-Mouse IgG (H + L), 1:10,000 (926-3221, LI-COR Biotechnology, Cambridge, UK) and/or IRDye 680LT Goat anti-Rabbit IgG (H + L), 1:40,000 (926-68023, LI-COR Biotechnology, Cambridge, UK), and visualized by an Odyssey infrared imager (LI-COR Biosciences). Data from densitometric analysis were background corrected.

**Immunofluorescence microscopy.** M1 and M2 ($1 \times 10^6$ cells) were seeded onto glass coverslips in a 12-well plate and cultured for 48 h. E. coli (ratio 1:50, macrophages: E. coli) or vehicle (PBS) was added at 37 °C and stopped after the indicated times by fixation with 4% paraformaldehyde solution. Acetone (3 min, 4 °C) followed by 0.25% Triton X-100 for 10 min at RT was used for permeabilization before blocking with normal goat serum 10% (50062Z, ThermoFisher). Samples were incubated with mouse monoclonal anti-5-LOX antibody, 1:100 (6A12 AB, 250 μg/ml; kindly provided by Dr. Dieter Steinhilber, Goethe-University-Frankfurt, Frankfurt, Germany)[16] and rabbit polyclonal anti-FLAP antibody, 5 μg/ml (ab85227, Abcam, Cambridge, UK Abcam), or mouse monoclonal anti-15-LOX-1 antibody, 1:100 (ab119774, Abcam, Cambridge, UK) and rabbit anti-5-LOX antibody, 1:100 (1550 AK6, kindly provided by Dr. Olof Radmark, Karolinska Institutet, Stockholm, Sweden) at 4 °C overnight. 5-LOX, 15-LOX-1, and FLAP were stained with the fluorophore-labeled secondary antibodies; Alexa Fluor 488 goat anti-rabbit IgG (H + L), 1:500 (A11034, ThermoFisher) and Alexa Fluor 555 goat anti-mouse IgG (H + L); 1:500 (A21424, ThermoFisher). Nuclear DNA was stained with ProLong Gold Antifade Mountant with DAPI (15395816, ThermoFisher). Samples were analyzed by a Zeiss Axiovert 200M microscope, and a Plan Neofluar ×40/1.30 Oil (DIC III) objective (Carl Zeiss, Jena, Germany). An AxioCam MR camera (Carl Zeiss) was used for image acquisition.

**Proximity ligation assay.** To detect in situ interaction of 5-LOX with FLAP in M1 and M2, an in situ PLA was performed as described[16]. Briefly, cells were treated, fixed, and incubated with the primary antibodies as described for immunofluorescence microscopy. The cells were incubated with species-specific secondary antibodies conjugated with oligonucleotides (Duolink In Situ PLA probe anti-mouse MINUS, DUO92004 and anti-rabbit PLUS, DUO92002, Sigma, Taufkirchen, Germany) for 1 h at 37 °C. By addition of two other circle-forming DNA

oligonucleotides and a ligase (30 min at 37 °C), the antibody-bound oligonucleotides form a DNA circle when the target proteins are < 40 nm distant from each other. The newly generated DNA circle was amplified by rolling circle amplification and visualized by hybridization with fluorescently labeled oligonucleotides (Duolink In Situ Detection Reagents FarRed, DUO92013, Sigma, Taufkirchen, Germany). Nuclear DNA was stained with DAPI. The PLA interaction signal appears as a fluorescent spot (magenta) and was analyzed by fluorescence microscopy using a Zeiss Axiovert 200M microscope, and a Plan Neofluar ×40/1.30 Oil (DIC III) objective (Carl Zeiss, Jena, Germany).

**$Ca^{2+}$ imaging.** Adherent M1 and M2 ($5 \times 10^6$ cells) were pre-stained with Fura-2/AM (2 μM) for 45 min at 37 °C in the dark. After two washing steps, cells were resuspended in PG-BSA buffer (PBS, 0.1% glucose, 0.1% fatty acid-free BSA) at a density of $1.25 \times 10^6$/ml. In total, 200 μl of the cell suspension were transferred into a 96-well plate and 1 mM $CaCl_2$ was added. After 10 min, 1 μM fMLF, 2 μM ionomycin, E. coli (macrophages: E. coli 1:50) or vehicle (PG-BSA) was added. The signal was monitored in a thermally (37 °C) controlled NOVOstar microplate reader (BMG Labtechnologies GmbH, Offenburg, Germany; emission at 510 nm, excitation at 340 nm ($Ca^{2+}$-bound Fura-2) and 380 nm (free Fura-2)). After cell lysis with Triton X-100, the maximal fluorescence signals were monitored, and after chelating $Ca^{2+}$ with 20 mM EDTA, the minimal fluorescence signals were recorded. The ratio of signals obtained with Triton X-100 subtracted by the signals obtained at basal fluorescence intensity (shown as Δratio) of each experiment was set to 100%.

**Real-time imaging of phagocytosis by human macrophages.** M1 or M2 (50,000 cells/well in PBS + Ca/Mg) was plated onto 8-well chamber slides. Chamber slides were kept in a Stage Top Incubation system for microscopes equipped with a built-in digital gas mixer and temperature regulator (TOKAI HIT model INUF-K14). RvD5 or LTB$_4$ (10 nM) was added to the cells for 15 min, followed by addition of BacLight Green-labeled E. coli (macrophage: E. coli ratio = 1:50). Images were then acquired every 10 min for 3 h (37 °C) with Keyence BZ-9000 (BIOREVO) inverted fluorescence phase-contrast microscope (×20 objective) equipped with a monochrome/color switching camera using BZ-II Viewer software (Keyence, Itasca, IL, USA). Green fluorescence intensity was quantified using BZ-II Analyzer.

**Statistical analyses.** The sample size for experiments was chosen empirically based on previous studies to ensure adequate statistical power. Results are expressed as mean ± standard error of the mean (S.E.M.) of $n$ observations, where $n$ represents the number of experiments with cells from separate donors, performed on different days, as indicated. Analyses of data were conducted using GraphPad Prism 7 software (San Diego, CA). Two-tailed $t$ test was used for comparison of two groups. For multiple comparison, one-way analysis of variance (ANOVA) with Bonferroni or Dunnett´s post hoc tests were applied. The criterion for statistical significance is $P < 0.05$. PCA was carried out using SIMCA 13.0.3 software (MKS Data Analysis Solutions).

**Data availability.** The data that support the findings of this study are available from the corresponding authors upon request.

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

## Acknowledgements

This work was supported by the Deutsche Forschungsgemeinschaft (SFB1127, Chem-BioSys) and in part by Grants from National Institutes of Health (NIH) NIGMS P01GM095467. J.G. received a Carl-Zeiss stipend. These studies were initiated with O.W. while on sabbatical in the C.N.S. laboratory, Boston.

## Author contributions

O.W., J.G., S.L., X.D.l.R., and N.C. performed these experiments; O.W., J.G., S.L., X.D.l.R., M.W., N.C., and P.C.N. performed analyses; O.W. and C.N.S. designed the study, and all authors contributed to manuscript preparation.
