## [Peer Review File · Nature Communications]

Reviewers' comments:

Reviewer #1 (Remarks to the Author):

This is an interesting and well-written and presented manuscript. The authors have reported that bacteria activate human macrophages to differentially produce specific resolving or eicosanoid signals and clearly shown that macrophages respond to these bacteria differentially to stimulate key lipids that defines their phenotypes.

Suggested Revision Comments

Please indicate more clearly the novelty of the findings in the manuscript.

Please discuss the mechanism by which these pathogenic bacteria stimulate the different lipid synthesis pathways - e.g., TLR ligation, etc.

The observation that only pathogenic bacteria stimulate SPMs is particularly interesting. Was there any difference between the bacterial groups tested and LOX/FLAP expression or localization, which could account for the discrepancy?

Given that the major substrates (AA, DHA, EPA) are produced at similar levels between M1 and M2 macrophages in this study, the difference in LM production is likely not substrate-limited. Do the authors suspect that LOX expression, LOX trafficking, or LM receptor expression is the limiting factor and/or major control point for SPM production and downstream effects, as all of these areas were shown to be altered in this study?

If M2 macrophages are the major producers of SPMs, why do M1 macrophages have higher expression of DRV1 and demonstrate a greater responsiveness to SPMs in terms of phagocytic increase? Do these M1 macrophages have greater bacterial killing abilities?

The discussion of 5-LOX and 15-LOX localization is merely a recap of the results- the implications of a lack of co-localization and how this contributes to LM regulation should be included.

Minor Points

Perhaps FMLP should be changed to FMLF throughout as FMLF is the correct nomenclature.

p.10, the sentence beginning "Polarization towards M1 gave consistent..." is not clear, particularly regarding the "temporal rather constant" phrase.

There are some minor grammatical errors (e.g., using significant instead of significantly, "did not increased," etc.), please correct.

Please verify which form of E.coli is being used throughout the methods.

Reviewer #2 (Remarks to the Author):

In the present report, authors deciphered that several lipid mediators are produced in a different way between M1 and M2 macrophages stimulated with E. coli and S. aureus. The different pattern and temporal production are supported by the expression and distribution of enzymes involved in their syntheses. The findings provide a novel insight on the molecular mechanism how M1 and M2 macrophages exhibit different phenotypes, at least partly. Pro-resolving functions of lipid mediators derived from AA, EPA and DHA were proposed by Serhan's group, and now the concept is broadly accepted by a number of in vivo and in vitro experiments. The present study adds the

strong biochemical mechanism by combining macrophage subtypes and lipid mediator profiling. Especially, it is an intriguing finding that LTB₄ acts at the initiation whereas RvD5 formed by M2 acts as a resolution signals to M1. The experiment is well executed and description, statistical analyses and ethical comments are appropriate. Several issues to be addressed to strengthen the study.

1. The study will provide much impact if the mouse *in vivo* studies are added; for example; inoculation of bacteria to the mice, and determine the temporal and spatial distribution of M1 and M2 macrophages and lipid mediators together with histological data. It is nicer to examine these effects by gene knockout mice (either receptor, or enzymes). Some experiments are necessary to validate *in vivo* relevance of the present cell assays.
2. Figure 4 shows the difference between pathogenic *E. coli* and their attenuated mutants in producing SPM and LTB₄, but not PGE₂. The control experiments by LPS-stimulation in parallel will help understanding.
3. Figure 2. What is the mechanism of 15-Lox induction by *E. coli*. The absence or weak expression of FLAP supports the little formation of LTB₄ in M2 macrophages. However, even in the absence (or weak expression) of FLAP, are 5-, 15-DHETE or other resolving produced in M2 macrophages? What is the role of FLAP?
4. Figure 2B shows an enough amount of PGD₂ in M2 macrophages, while PGD₂ seems not produced in M2 (AA metabolome) in Fig. 3C. How do authors explain the discrepancy?
5. Figure 5 shows nice staining of 5-LO and 15-LOX-1 in macrophages. 5-Lox is present in nucleus (and less in the cytosol) and causes translocation to nuclear membrane around 60-90 min after *E. coli* infection. Is the translocation dependent on Ca mobilization and/or phosphorylation? 15-Lox in contrast distributed rather unevenly in the cytosol (Fig. 5C). Which organellae or vacuoles contain 15-Lox1?

Reviewer #1 (Remarks to the Author):

This is an interesting and well-written and presented manuscript. The authors have reported that bacteria activate human macrophages to differentially produce specific resolving or eicosanoid signals and clearly shown that macrophages respond to these bacteria differentially to stimulate key lipids that defines their phenotypes.

1. Please indicate more clearly the novelty of the findings in the manuscript.

Authors: Thank you for pointing this out. We now more clearly indicate the novelty of these findings in the introduction (page 4, bottom line to page 5) and discussion (bottom paragraph page 11 to page 12; page 13, 2nd paragraph) as well as re-stated the title. In short, this is the first set of results demonstrating that human macrophages biosynthesize different profiles of lipid mediators when they encounter pathogenic bacteria. They switch from producing LTB₄ to resolvin D5. The resolvin D5 then functions in enhancing bacterial phagocytosis with these cells.

2. Please discuss the mechanism by which these pathogenic bacteria stimulate the different lipid synthesis pathways - e.g., TLR ligation, etc.

Authors: This is now discussed on Page 14, lines 10-21; page 12, lines 2-4. Briefly, the pathogenic bacteria studied here activate LOX and COX pathways, while non-pathogenic *E. coli* (attenuated or different strains) mainly activate COX with only low amounts of LOX pathway products. TLR4/CD14 activation in either M1 or M2 using lipopolysaccharide (LPS) did not substantially activate the biosynthesis of LOX pathway products (see Suppl. Fig. 2, and the new Fig. 4). Cell activation by LPS also failed to stimulate translocation of LOX (new Supplementary Fig. 4), but did stimulate PG formation, especially in the M1 cells (Supplementary Fig. 2).

Pathogenic bacteria secrete pore-forming toxins (e.g. α -hemolysin) that can activate macrophages by elevation of intracellular Ca²⁺ and phosphorylation events (e.g. p38 MAPK). Along these lines, we carried out additional experiments and added new results which indicate that the pathogenic *E. coli* (i) elevate intracellular Ca²⁺ in both M1 and M2 (new Figure 6A) and (ii) in the absence of Ca²⁺ (chelation by EDTA plus BAPTA/AM) fail to induce 5-LOX/15-LOX translocation (new Fig. 6B) and LM formation (new Supplementary Fig. 6) in macrophages. These results sharply contrast with those obtained with non-pathogenic *E. coli*.

3. The observation that only pathogenic bacteria stimulate SPMs is particularly interesting. Was there any difference between the bacterial groups tested and LOX/FLAP expression or localization, which could account for the discrepancy?

Authors: The responses with *S. aureus* and those with pathogenic *E. coli* on LOX product formation in M1 and M2 cells were quite similar. These were clearly distinct from the small effects of non-pathogenic *E. coli* strains JM109 and BL21 or attenuated *E. coli*, where only PGs were formed when either M1 or M2 macrophages were exposed to the non-pathogenic bacteria.

Since bacteria-activated SPM formation in either M1 or M2 cells occurred within 30 - 90 min, it appears that modulation of LOX/FLAP at the expression level is negligible in these cell phenotypes.

We agree with the reviewer that LOX localization may determine this difference. Along these lines, non-pathogenic *E. coli* (BL21) failed to active LOX product formation (please see the new results in Supplementary Fig. 6) and did not cause translocation of either 5-LOX or 15-LOX (see new Fig. 6B).

4. Given that the major substrates (AA, DHA, EPA) are produced at similar levels between M1 and M2 macrophages in this study, the difference in LM production is likely not substrate-limited. Do the authors suspect that LOX expression, LOX trafficking, or LM receptor expression is the limiting factor and/or major control point for SPM production and downstream effects, as all of these areas were shown to be altered in this study?

Authors: Thank you. We now discuss this important question in light of our results (page 12, second paragraph; page 15, line 3-6). The >50-fold higher 15-LOX-1 protein expression in M2 versus M1 cells certainly can account for the greater SPM production by these cells, while the >4-fold higher FLAP expression in M1 may account for greater LTB₄ formation in this phenotype. Also, 15-LOX-1 trafficking and access towards substrates (AA, DHA, EPA) at a distinct locale might be a determinant. We hypothesize that defined subcellular access of substrate for 5-LOX or 15-LOX-1 at different sites in the two phenotypes contributes to the differences in LM production. We have no experimental evidence that different LM receptor expression control differential LM production. These are of interest and will be a subject for future studies but appear to be outside of the scope of the present manuscript.

5. If M2 macrophages are the major producers of SPMs, why do M1 macrophages have higher expression of DRV1 and demonstrate a greater responsiveness to SPMs in terms of phagocytic increase? Do these M1 macrophages have greater bacterial killing abilities?

Authors: We agree that this is an important point and we have carried out additional experiments. We measured bacterial killing by both M1 and M2 cells. In 2-hour incubations with *E. coli*, we found that M2 have higher killing abilities than M1 (page 11, lines 3-7). We suggest that M2 produce SPM during their exposure to *E. coli* that may increase their bacterial killing capacities as compared to M1 that hardly produce SPM. For direct comparison, RvD5 stimulated M1 phagocytosis to a greater extent than M2, probably because of higher DRV1 expression (see Fig. 7B). *In vivo*, SPM produced by M2 may be utilized by M1 to increase the bacterial killing abilities. These new results and suggestions appear on pages 11 and 13 in the revised manuscript.

6. The discussion of 5-LOX and 15-LOX localization is merely a recap of the results-the implications of a lack of co-localization and how this contributes to LM regulation should be included.

Authors: Thank you. This point is now discussed in more detail (page 15) and the overall discussion is shortened.

Minor Points

Perhaps FMLP should be changed to FMLF throughout as FMLF is the correct nomenclature.

Authors: Done.

p.10, the sentence beginning "Polarization towards M1 gave consistent..." is not clear, particularly regarding the "temporal rather constant" phrase.

Authors: Done, rephrased.

There are some minor grammatical errors (e.g., using significant instead of significantly, "did not increased," etc.), please correct.

Authors: Done.

Please verify which form of E.coli is being used throughout the methods.

Authors: Done.

Reviewer #2 (Remarks to the Author):

In the present report, authors deciphered that several lipid mediators are produced in a different way between M1 and M2 macrophages stimulated with E. coli and S. aureus. The different pattern and temporal production are supported by the expression and distribution of enzymes involved in their syntheses. The findings provide a novel insight on the molecular mechanism how M1 and M2 macrophages exhibit different phenotypes, at least partly. Pro-resolving functions of lipid mediators derived from AA, EPA and DHA were proposed by Serhan's group, and now the concept is broadly accepted by a number of in vivo and in vitro experiments. The present study adds the strong biochemical mechanism by combining macrophage subtypes and lipid mediator profiling. Especially, it is an intriguing finding that LTB4 acts at the initiation whereas RvD5 formed by M2 acts as a resolution signals to M1. The experiment is well executed and description, statistical analyses and ethical comments are appropriate. Several issues to be addressed to strengthen the study.

1. The study will provide much impact if the mouse in vivo studies are added; for example; inoculation of bacteria to the mice, and determine the temporal and spatial distribution of M1 and M2 macrophages and lipid mediators together with histological data. It is nicer to examine these effects by gene knockout mice (either receptor, or enzymes). Some experiments are necessary to validate in vivo relevance of the present cell assays.

Authors: As stated above in a phone call with the Associate Editor Dr. Nicholas Bernard on June 12, 2017, we discussed and he agreed that this manuscript should focus solely on the human results for the two phenotypes and bacteria, rather than trying to also include mouse experiments in this manuscript. Temporal formation of lipid mediators with endogenous SPM production after inoculation of bacteria in mice was reported earlier by our group, see Chiang N. et al. (2012) *Nature*. The focus of this present contribution is on human macrophages and lipid mediators; the situation in mice is quite different. For example, the human 15-LOX-1 in leukocytes that contributes to the biosynthesis of many of the SPM is a different enzyme from the mouse homolog that is a combined 12/15-LOX. Hence, the activation of these pathways in mice cells and humans for SPM production are not orthogonal and are controlled by different mechanisms at the intracellular level.

2. Figure 4 shows the difference between pathogenic *E. coli* and their attenuated mutants in producing SPM and LTB₄, but not PGE₂. The control experiments by LPS-stimulation in parallel will help understanding.

Authors: Thank you for this point. We carried out additional experiments to address this and now added the requested results (LM formation in LPS-fMLP-stimulated M2) to the revised Fig. 4; see also LM formation and LOX translocation in LPS-fMLP-stimulated M1 in Supplementary Figures 2 and 4).

3. Figure 2. What is the mechanism of 15-Lox induction by *E. coli*. The absence or weak expression of FLAP supports the little formation of LTB₄ in M2 macrophages. However, even in the absence (or weak expression) of FLAP, are 5, 15-DHETE or other resolving produced in M2 macrophages? What is the role of FLAP?

Authors: Thank you for raising this point; see also comments 2., 3. and 4. from Reviewer #1 and our respective responses. In contrast to 5-LOX, little is known about cellular activation of 15-LOX-1 in human macrophages with bacteria. We now provide results, which indicate that Ca²⁺ signaling is essential for activation of 15-LOX-1 and SPM production in M2 (new Fig. 6A,B and new Supplementary Fig. 6). In addition, we also added new results using a FLAP inhibitor which does not appear to impact 15-LOX-1-mediated SPM formation in the M2 macrophages (Supplementary Fig. 5). These new results indicate that FLAP does not play a prominent role of in RvD5 formation (see, page 10, 1st paragraph).

4. Figure 2B shows an enough amount of PGD₂ in M2 macrophages, while PGD₂ seems not produced in M2 (AA metabolome) in Fig. 3C. How do authors explain the discrepancy?

Authors: Thank you for this point, we clarified it (page 7, line 5-3 from bottom). We used different scales of the Y-axes in Fig. 2B (showing PGD₂ after 48 hrs polarization) and Fig. 3C (showing PGD₂ after during the entire time course from 0 to 72 hrs) of M1/M2 polarization. The PGD₂ levels in M1 (172.7 pg) and M2 (379.2 pg) shown in Fig. 2B are recorded in cells that had been polarized for 48 hrs (see also Suppl. Table 2). These results correspond to the values reported in Fig. 3C at 48 hrs. Due to the peak size of PGD₂ in this chromatographic system after 6 hrs (approx. 3000 pg) in M1 that leads to wide scaling of the Y-axis in Fig. 3C, the PGD₂ levels at 48 hrs appear very low in M2 and were difficult to inspect.

5. Figure 5 shows nice staining of 5-LO and 15-LOX-1 in macrophages. 5-Lox is present in nucleus (and less in the cytosol) and causes translocation to nuclear membrane around 60-90 min after *E. coli* infection. Is the translocation dependent on Ca mobilization and/or phosphorylation? 15-Lox in contrast distributed rather unevenly in the cytosol (Fig. 5C). Which organelles or vacuoles contain 15-Lox1?

Authors: See comments above to Points 2 – 4 from Reviewer #1. Yes, the *E. coli*-induced translocation of 5-LOX and 15-LOX-1 is Ca²⁺-dependent (new Fig. 6B) and *E. coli* elevate intracellular Ca²⁺ levels in M1 and M2 (new Fig. 6A); this has now also been discussed on page 12, line 2-4 and page 15.

Phosphorylation of 15-LOX-1 has not been reported yet but might be a potential additional mechanism of how *E.coli* could activate this enzyme. We are currently investigating the subcellular locale of 15-LOX-1 translocation (e.g. organelles or vacuoles) by electron microscopy and the role of phosphorylation of 5-LOX and of 15-LOX-1 for their translocation/activation in more detail. However, these studies require further experiments that cannot be performed within this revision and we feel are outside of the present scope of this manuscript.

We thank the reviewers and editor for their time and very helpful comments on this manuscript. I trust that the revised manuscript is improved and now suitable for publication in Nature Communications. Looking forward to hearing from your office.

Sincerely,

Charles N Serhan

and

Oliver Wertz

/mhs

Reviewers' comments:

Reviewer #1 (Remarks to the Author):

The authors should be congratulated on their efforts to address all the points raised by the reviewers. They have included new data and amended the manuscript appropriately.

Reviewer #2 (Remarks to the Author):

It appears that authors addressed adequately most comments by the reviewer with additional experiments. There are a number of issues still to be clarified. One critical point still unclear to the reviewer (and possibly to editors and readers) is the role(s) of FLAP in lipid mediator formation. According to the present study, FLAP is one of the major discriminators between M1 and M2 macrophages, and of molecular bases of different lipid mediator productions between 2 cell types.

In supplementary figure 5, they showed effects of FLAP inhibitor on the change of lipid mediator formation using M2 macrophages.

1. Why the amount of RvD5, 7,14S-dHETE, Mar1 are not changed or elevated, in spite that they all require 5-lipoxygenase reaction?
2. The similar experiments have to be done for M1 macrophages, as M1 expresses higher FLAP than M2.
3. It is not clear how AA, EPA levels are decreased.

Reviewer #1 (Remarks to the Author):

The authors should be congratulated on their efforts to address all the points raised by the reviewers. They have included new data and amended the manuscript appropriately.

Response: We thank the reviewer for their helpful and constructive comments on our manuscript. These statements are much appreciated.

Reviewer #2 (Remarks to the Author):

It appears that authors addressed adequately most comments by the reviewer with additional experiments. There are a number of issues still to be clarified. One critical point still unclear to the reviewer (and possibly to editors and readers) is the role(s) of FLAP in lipid mediator formation. According to the present study, FLAP is one of the major discriminators between M1 and M2 macrophages, and of molecular bases of different lipid mediator productions between 2 cell types.

In supplementary figure 5, they showed effects of FLAP inhibitor on the change of lipid mediator formation using M2 macrophages.

Response: We thank the reviewer for critical evaluation of our revised manuscript. We have clarified the points raised below and revised our manuscript accordingly. Specific responses are as follows.

1. Why the amount of RvD5, 7,14S-dHETE, MaR1 are not changed or elevated, in spite that they all require 5-lipoxygenase reaction?

Response: Thank you for raising this point. We have clarified this in the revised manuscript on page 10 and have included the following with appropriate references:

“MaR1 and 7S,14S-diHDHA from the maresin pathway are biosynthesized via initial oxygenation by the human 12-LOX and their biosynthesis is independent of 5-LOX^{17,18}. Also, PD1 biosynthesis is initiated via the 15-LOX-1 and does not require 5-LOX^{3,19}. Thus, FLAP is not involved in the biosynthesis of PD1, MaR1 or 7S, 14S-diHDHA and MK886 did not impact the production of their pathway markers (i.e. 17-HDHA, 14-HDHA).”

2. The similar experiments have to be done for M1 macrophages, as M1 expresses higher FLAP than M2.

Response: Thank you for raising this point. This is certainly a logical conclusion; however, given the very low amounts of SPM produced by M1 macrophages, an accurate quantitative parallel assessment is not feasible. Thus, we have revised our manuscript to clarify this point as follows on page 13:

“The M1 cells express more FLAP than M2. The 15-LOX-1 is essentially absent in M1 and thus SPM (i.e., RvD5, 7S,14S-diHDHA, Mar1) formation is very low just above the detection limits with levels that were ~ 100-fold lower than in the M2 cells. This made it not plausible to accurately assess a role for FLAP (using MK886) in SPM formation in M1 cells since they did not produce appreciable amounts of SPMs.”

We have also revised the Discussion on page 16:

“It is possible that in M2 macrophages the RvD5 is produced via 15-LOX-1 alone to insert both molecules of molecular oxygen without the need of 5-LOX or FLAP. Formation of the two precursors of RvD5 (7S,17S-diHDHA), namely 7S-HDHA and 17S-HDHA, was not directly inhibited by MK886 in M2 cells suggesting that the oxygenations at 7- and 17-positions in these cells from endogenous substrate leading to RvD5 appear to be 5-LOX/FLAP-independent. Along these lines, 15-LOX-1 is able to catalyze all steps in the biosynthesis of LXA₄, without the need of 5-LOX²⁷. In this context, it was shown that activated cytoplasmic 5-LOX in macrophages, without access to FLAP favors SPM biosynthesis over the production of LTB₄²⁸, suggesting a limited role for FLAP in leukotriene biosynthesis.”

3. It is not clear how AA, EPA levels are decreased.

Response: Thank you for raising this point. We have addressed this and clarified our manuscript on page 15 as follows:

“Apart from substrate utilization of AA, and EPA, we cannot fully explain all of the effects of MK886 on such complex lipid mediator networks. Some actions of MK886 may be off target in these cell types. Also, cPLA₂ (that releases AA and EPA) may participate within the 5-LOX/FLAP complex which is disrupted by MK886 and might thus negatively affect cPLA₂. It is possible that LTB₄ or 5-HETE may act as autocrine enhancers for further AA and EPA release that would be abolished when MK886 blocks LTB₄ and 5-HETE biosynthesis.”

In closing, we thank the reviewers for their very helpful and insightful comments that have helped to improve the presentation of these results and our manuscript.

Sincerely,

Charles N Serhan

REVIEWERS' COMMENTS:

Reviewer #2 (Remarks to the Author):

Author addressed adequately to comments by the reviewer. The manuscript is now acceptable for publication. The reviewer apologizes authors, as I did not see these previous references, and made misunderstandings.

Takao Shimizu